# Influence of Age and Immunostimulation on the Level of Toll-Like Receptor Gene (*TLR3*, *4*, and *7*) Expression in Foals

**DOI:** 10.3390/ani10111966

**Published:** 2020-10-26

**Authors:** Anna Migdał, Łukasz Migdał, Maria Oczkowicz, Adam Okólski, Anna Chełmońska-Soyta

**Affiliations:** 1Department of Genetics, Animal Breeding and Ethology, Faculty of Animal Sciences, University of Agriculture in Krakow, al. 29 Listopada 46, 31-425 Kraków, Poland; lukasz.migdal@urk.edu.pl; 2Department of Animal Molecular Biology, National Research Institute of Animal Production, Krakowska 1, 32-083 Balice, Poland; maria.oczkowicz@izoo.krakow.pl; 3Institute of Veterinary Science, University Centre of Veterinary Medicine UJ-UR, University of Agriculture in Krakow, al. Mickiewicza 24/28, 30-059 Kraków, Poland; adam.okolski@urk.edu.pl; 4Laboratory of Reproductive Immunology, Hirszfeld Institute of Immunology and Experimental Therapy, Polish Academy of Sciences, Weigla 12 Street, 53-114 Wroclaw, Poland; anna.chelmonska-soyta@hirszfeld.pl; 5Department of Immunology, Pathophysiology and Veterinary Preventive Medicine, Division of Immunology and Veterinary Preventive Medicine, Faculty of Veterinary Medicine, Wroclaw University of Environmental and Life Sciences, Norwida 31 Street, 50-375 Wroclaw, Poland

**Keywords:** *TLR3*, *TLR4*, *TLR7*, foals, immunostimulation, gene expression

## Abstract

**Simple Summary:**

Detailed knowledge of the molecular mechanisms of immunoglobulin synthesis appears necessary for a better understanding of foal immunity maturity and its influencing factors. At the same time, it encourages studies regarding the influence of the signaling cascade’s proteins on the primary immunological response, which provides an opportunity to develop extremely precise methods of regulating acquired immunity. The results revealed that the expression of the*TLR3* and *TLR4* genes, as well as the levels of immunoglobulins and interleukins, can be modulated by stimulation with the pharmacological agent, and that the expression of the *TLR3* and *TLR4*genes in peripheral blood cells is dependent on age.

**Abstract:**

The aim of this study was to investigate the molecular mechanisms leading to the identification of pathogens by congenital immune receptors in foals up to 60 days of age. The study was conducted on 16 foal Polish Pony Horses (Polish Konik) divided into two study groups: control (*n* = 9) and experimental (*n* = 7). Foals from the experimental group received an intramuscular duplicate injection of 5 mL of Biotropina (Biowet) at 35 and 40 days of age. The RNA isolated from venous blood was used to evaluate the expression of the*TLR3*, *TLR4*, and *TLR7* genes using RT-PCR. The results of the experiment demonstrated a statistically significant increase in the level of *TLR3* gene expression and a decrease in the level of*TLR4* gene expression with foal aging. The level of *TLR7* gene expression did not show age dependence. Immunostimulation with Biotropina had a significant impact on the level of the genes’ expression for Toll-like receptors. It increased the level of *TLR4* expression and decreased *TLR3* expression. Thus, it was concluded that the expression of the*TLR3* and *TLR4*genes in peripheral blood cells is dependent on age. This experiment demonstrated a strong negative correlation between *TLR3* and *TLR4* gene expression.

## 1. Introduction

Immune response differs in newborn and adult horses. Despite involving similar components, the regulation of immunity and the response to antigens vary. Foals are born with small, non-protective amounts of endogenous serum immunoglobulins (i.e., IgM and IgG) [1]. Immunoglobulin transport is limited due to the structure of the horse placenta (placenta spuria). That is why the suckling of colostrum is essential in the first hours of a foal’s life. Immunological outcomes in newborn foals differ as compared to adults and are distinguished by modified cytokine profiles, as well as reduced antibody and T-cell responses [2]. Moreover, foals have a very low level of immunoglobulins in their blood plasma. An innate immune system composed of pre-existing or rapidly induced defenses is critical for newborn foals, while an antigen-specific response requires exposure to pathogens and time for development after birth [3]. The effectiveness of the immunological reaction is controlled by a complexity of direct and indirect mechanisms involving interactions among various cells and cytokine-induced actions. Many different immune cells are involved in maintaining a balanced immune response [4]. Receptors called pathogen recognition receptors (PRRs) represent very important elements found in immune cells. Thanks to their conservative structure, these receptors can recognize signals associated with a variety of pathogens. Pathogen-associated molecular patterns (PAMPs), PRR ligands, are molecules specific to viruses, bacteria, and other microorganisms with evolutionarily conserved structures [5]. Toll-like receptors (TLRs) are the most closely investigated PRRs and are one of the most essential components of immune responses [6,7]. Toll-like receptors play a key role in activating and stimulating innate as well as acquired immunity [8]. Identification of the threat and activation of TLRs triggers an immune response, leading to the elimination of this threat from the organism, which involves two basic reactions, namely, inflammatory, and antiviral. Cells with activated receptors release large amounts of proinflammatory cytokines, chemokines, and defensins, and these released factors initiate the migration and aggregation of immune cells (e.g., leukocytes, macrophages, mast cells, and dendritic cells) at the site of the pathogen invasion [7]. Activated TLRs present on the surface of macrophages lead to increased synthesis of the proinflammatory cytokines IL-1, -6, -8, -12, and TNF-α. In addition, complexes of ligands and TLR4 receptors increase the phagocytic activity of macrophages and stimulate the production of reactive oxygen species (ROIs) and the synthesis of nitric oxide (NO). TLR-activated macrophages enhance the expression of major histocompatibility complexes I and II (MHCI and MHCII), CD80, CD86, and co-stimulators that make immune cells more efficient in displaying T-cell antigens that induce specific immune responses [9,10].

Approximately 20% of foals die before the end of the second month, which brings significant economic and breeding losses [11,12]. This justifies undertaking research into new and novel techniques for stimulation of the immune system. Gaining insight into the behavior of TLRs seems to be necessary for expanding our understanding of the mechanism responsible for the development of foal resistance/immunity, and the identification of determinants for further building it up. In addition, it is also an important element contributing to finding innovative solutions in the fight against infections and new ways to improve the prevention and treatment of infections in animals. The paradox of neonatal vaccination is the need of immediate protection during early days, the perceived limitations of the immune system of neonate foals, and the theory of maternal antibody interference [13]. Studies have shown that the immune system of neonatal foals is also naive and immature relative to juvenile and adult horses [14,15,16,17]. Several studies have suggested that basal TLR expression in full-term neonatal blood monocytes is similar to that of adults [18,19]. The TLR-mediated production of cytokines by neonatal monocytes, however, is very different in newborns compared to that of adults [19]. Thus far, little is known about the development of the horse immune system during pre- and postnatal periods, which negatively affects the ability to devise strategies for maintaining and improving foal health. Based on its biological properties, as well as the influence of Toll-like receptors on the immune response traits of farm animals and humans, we hypothesized that gene expression for Toll-like receptors TLR-3, TLR-4, and TLR-7 in foals is dependent on factors such as age and immunostimulation. The aim of this work was to investigate the molecular mechanisms leading to the identification of pathogens by congenital immune receptors in foals up to 60 days of age, including the verification of the hypothesis concerning age-related expression of the*TLR3*, *TLR4*, and *TLR7* genes.

## 2. Materials and Methods

This experiment was granted permission from the Local Ethics Committee in Kraków (no 37, 30 May 2016).

### 2.1. Animals and Feeding

Studies were carried out on 16 foals representing Polish Pony horses (Polish Konik). This primitive horse breed is genetically and phenotypically closely related to its wild ancestor, the Tarpan Horse (Eurasian wild horse) [20].

All foals with mares were kept in the same stable in individual boxes (size 2.15 × 3.50 m) on permanent straw bedding at the Experimental Station of the University of Agriculture in Krakow. All animals were clinically healthy throughout the experimental period. Mares of 5–17 years of age and 270–340 kg live body weight were not vaccinated during pregnancy. Foal birth weight was 27–35 kg, and weight loss on the first day of life was <1.5%. The horses had all been used by university students in the teaching program. No horses were used for equestrian purposes. Inclusion criteria consisted of foals born from healthy mares with no placentitis, a normal gestational period, an uneventful birth, and normal physical and neurological examination findings. The foals had to successfully stand and nurse within 2 h of birth and remain clinically healthy during the study period.

Mares were fed ad libitum with hay (*Lolium* 40% and *Trifolium L.* 20%) with the addition of oats in the amount of 1.5 kg/mare/day [21]. Foals were fed only with colostrum and mother’s milk ad libitum, without additional supplementation. Water was offered from automatic water drinkers (flow ~ 10 L/min).

### 2.2. Experimental Design

Two weeks before delivery, birth alarms (Abfohlsystem, Jan Wolters, Steinfeld, Germany) were placed in the labia, and mares were moved to box stalls inside a stable lit with natural light (Appendix A). During the experiment, foals were kept with their mothers in individual boxes, and when leaving the stalls with their mothers for the pasture, they were randomly assigned into the following groups:The control group (Group C) (*n* = 9)—foals without any pharmacological and feed additives that may influence immune system;The experimental group (Group E) (*n* = 7)—foals that were administered an immunostimulating agent.

For the immunostimulation, a commercially available immunostimulator was used in the present study, namely, Biotropine (Biowet Drwalew S.A., Drwalew, Poland), which consists of a mixture of inactivated Gram-positive bacteria, e.g., *Staphylococcus aureus* (74 mg/mL), *Streptococcus zooepidermicus* (24.6 mg/mL), *Streptococcus equi* (24.6 mg/mL), *Streptococcus equisimilis* (24.6 mg/mL), *Streptococcus agalactiae* (24.6 mg/mL), *Streptococcus dysgalactiae* (24.6 mg/mL), *Erysipelothrix insidiosa* (49 mg/mL),and Gram-negative bacteria, e.g., *Escherichia coli* (123 mg/mL) and *Pasteurella multocida* (123 mg/mL) as well as pork spleen extract (10 mg/mL). On days 35 and 40 after birth, the foals from the experimental group received an intramuscular (*m. pectoralis descendens*) injection of 5 mL of Biotropine.

### 2.3. Blood Sampling and Blood Analysis

Blood samples were collected from foals by jugular venipuncture. Blood samples were obtained from foals up until 60 days of age according to the following scheme: After birth before the first suckling and then on the 1st, 3rd, 5th, 10th, 20th, 30th, 40th, 50th, and 60th days of age. Three milliliters of blood were collected into TEMPUS tubes (Applied Biosystems, Foster City, CA, USA) with RNA stabilizing factor. Samples were stored at −20 °C until further processing. Isolation of RNA was carried out using TEMPUS SPIN (Ambion, Waltham, MA, USA) according to the manufacturer’s protocol (Appendix A). One microgram of RNA was transcribed into cDNA using a High-Capacity cDNA Reverse Transcription Kit (Applied Biosystems, Foster City, CA, USA) according to the manufacturer’s protocol.

A “No-RT” (non-reverse transcriptase) control was used for selected RNA samples to analyzed contamination in samples.

Gene expression analyses (Appendix A presents the reaction efficiency of each gene) were performed on an Illumina Eco system (Illumina, San Diego, CA, USA, Country)using TaqMan^®^MGB (Applied Biosystems, Foster City, CA, USA) probes (Table 1). Every sample was analyzed in triplicate in a final volume of 10 µL (Appendix A). Amplification was performed according to the following protocol: polymerase activation at 95 °C (2 min) and 40 cycles at 95 °C for 15 s and 60 °C for 1 min. The *SDHA* and *HPRT* genes were used as housekeeping genes (Table 1).

In addition, an analysis of the blood morphotic parameters was performed (Appendix A).

### 2.4. Statistical Analysis

Data are presented as means ± standard error. The data were analyzed using SAS 9.4 software (SAS Institute Inc., Cary, NC, USA). The Shapiro–Wilk test was considered the best test to check the normality of the distribution of random variables. Because the data did not have a normal distribution, the Kruskal–Wallis test was used with immunostimulation and age as the effects. The degree of association between the parameters was examined using a non-parametric Spearman’s rank correlation coefficient. Values ranging from 0.0 to 0.5, from 0.5 to 1.0, from −0.5 to 0.0, and from −1.0 to −0.5 indicate weak positive, strong positive, weak negative, and strong negative correlations, respectively.

## 3. Results

### 3.1. Influence of Age on the Expression of TLR3, 4, and 7 mRNA

The lowest expression of *TLR3* was observed during delivery (6.20 ± 0.89) (Figure 1—data presented from control group). After delivery, the level of *TLR3* mRNA increased. In the period between delivery and 60 days of age, the level of *TLR3* expression increased by 94.34%. We found a highly statistically significant difference between the age and expression of *TLR3* mRNA (*p* > 0.01), as presented in Table 2. From the 5th day of age, we found statistical differences between samples, with the highest expression level on the 60th day after delivery.

The highest level of *TLR4* mRNA expression was observed during the delivery (18.3 ± 2.6) of newborn foals. From the day of the delivery to 20th day of age, we observed a steady significant decrease of *TLR4* mRNA expression (Figure 1) in the blood of the examined foals. During the subsequent days of observation, the expression of the mRNA of this receptor remained at a similar level. The lowest expression value was observed at the 60th day of age (5.82 ± 0.96) (Table 2). Between the delivery day and the 60th day of age, the expression of mRNA for *TLR4* decreased by 76.89%. Statistical analysis showed highly statistically significant differences between the age of foals and the expression of *TLR4* mRNA (*p* < 0.01).

The expression ofthe*TLR7* gene remained statistically unchanged throughout the experiment (Figure 1). The highest values were observed during the day of delivery (9.20 ± 1.20), while the lowest was observed 20 days after delivery (7.20 ± 0.61). There was no statistically significant correlation between the age and the expression of *TLR7* mRNA (*p* > 0.2366) (Table 2).

### 3.2. Influence of Stimulation with Biotropina on the Expression of TLR3, 4, and 7 mRNA

Analysis of changes in the expression of *TLR3* mRNA after the injection of the immunostimulant (Group E) showed a decrease by 41.65% (Table 3), while in the control group (Group C), a dynamic increase in the expression was observed until the last day of observation (116.22 ± 13.93). Highly statistically significant differences were found between immunostimulation and the expression of *TLR3* mRNA.

The level of *TLR4* mRNA expression at 30 days of age was similar in both groups (Table 3). After Biotropina injection, *TLR4* mRNA expression increased by 41.33% (Group E), while the expression of *TLR4* mRNA in foals from Group C decreased by 20.22%. On the following days after immunostimulation, *TLR4* mRNA expression in the foals from Group E was higher, but we did not find statistically significant differences between groups. Statistical analysis showed statistically significant differences in *TLR4* mRNA expression after immunostimulation.

The initial expression of *TLR7* mRNA before first suckling was higher in Group C. In Group E, the highest level of expression was observed during delivery at 7.57 ± 0.88 (Table 3). In Group C, the expression was higher during the experiment compared to Group E, and we found highly statistically significant differences between groups (*p* < 0.001). No statistically significant differences were found between the expression of *TLR7* mRNA and immunostimulation.

The Spearman’s rank correlation test showed a strong negative correlation between the*TLR3* and *TLR4* genes, and lack of correlation between *TLR3* and *TLR7*as well as*TLR4* and *TLR7* (Table 4).

### 3.3. Influence of Age and Stimulation with Biotropina on the Level of Blood Morphotic Elements

Statistical analysis showed (Table 5):-A significant influence of age on the hematocrit level;-A highly significant influence of age on the hemoglobin level;-A significant influence of age on the level of erythrocytes;-A highly significant influence of age and immunostimulation on the level of leukocytes;-A highly significant influence of age and immunostimulation on the level of lymphocytes;-A highly significant influence of immunostimulation on the number of monocytes;-A significant influence of age and immunostimulation on the number of neutrophils.-A highly significant influence of immunostimulation, significant influence of age on the number of basophils;-A highly significant influence of immunostimulation and age on the number of basophils.

## 4. Discussion

Infectious diseases are common in foals between the first and fifth months of age. Analysis of the concentrations of immune system components during this period in healthy and infected foals may help understand the basics of the maturation of the immune system and also better understand infection mechanisms [22]. To the best of our knowledge, there are very limited reports where weekly collections and the expression of immune-related genes have been performed. Therefore, our results may be interesting for better understanding the changes during the first weeks of a foal’s life and the changes it undergoes during this time. Most studies report data from the first 24 h of a foal’s life, from the first 42 days, and very often from adult horses. Moreover, most data include thoroughbreds, while we performed our analysis on a primitive horse breed that is known for their adaptation to harsh conditions. While this study was performed on a primitive domestic breed, the results from this study may differ from potential results if performed on selectively breed domestic horses. Flaminio et al. [22] reported that healthy lymphocytes of healthy foals were the lowest at birth and that values increased until the sixth month of age. In our study, we obtained similar results; however, in Polish Pony, lymphocyte counts were higher (Table 5).

### 4.1. Changes in TLR4 Gene Expression

Because of the increased vulnerability of foals to some pathogens (e.g., *Rhodococcus equi*), it seems reasonable to analyze changes in the expression of the genes that are responsible for the recognition of the conserved constituents of pathogens [23]. In the present study, a highly significant influence of age and stimulation with Biotropina was observed for *TLR4*. Data available in the literature indicate that the influence of age on *TLR4* expression in horses is contradictory. Vendrig et al. [24] found no differences between the expression of *TLR4* in the blood mononuclear cells of foals at 12 h of life or in adult horses. Stimulation with lipopolysaccharides (LPS) resulted in higher *TLR4* mRNA expression in adult horses, while no response to LPS stimulation was found in foals in an in vivo study [24]. In contrast, Tessier et al. [25], having compared *TLR4* mRNA expression in umbilical cord blood and peripheral blood from adult horses, demonstrated higher*TLR4* mRNA expression in umbilical cord blood in response to LPS administration. The influence of age on the expression of *TLR4*was observed by Hansen et al. [26] in horses aged between 5 and 27 years. Higher *TLR4* mRNA levels were found in younger horses, but the decrease in mRNA levels with age were not statistically significant. *TLR4* mRNA levels were higher in blood mononuclear cells compared to the same cells from pulmonary vascular secretions. Osorio et al. [27] and Strong et al. [28] evaluated the expression of the *TLR4* gene in the first weeks of a calf’s life, and their results were similar to the one produced in our study. The highest level of*TLR4* gene expression was observed after birth, and a statistically significant decrease in expression was reported during the following days. A similar trend was identified in our study. Yerkovich et al. [29] and Levy et al. [30] showed that the expression of the *TLR4* gene was significantly higher in peripheral blood in premature and full-term infants than in adults, both before and after LPS stimulation [31]. This trend, indicating a decreasing expression of *TLR4* with age, was also found in humans and mice [32,33]. On the other hand, some results illustrate a higher expression of *TLR4* in newborns compared to adults [34] or decreasing expression of *TLR4* after stimulation [35]. The differences in *TLR*4 expression revealed in these studies and in our experiment may be caused by different concentrations of LPS in the stimulation.

### 4.2. Changes in TLR3 Gene Expression

We found that *TLR3* mRNA levels increased with the age of the foals. Our results are in agreement with other reports about horses and mostly human newborns [25,36,37,38]. Interestingly, there are reports proving epigenetics control *TLR3* expression mechanisms [36]. As Porras et al. [36] reported in their results from healthy donors, it can be presumed that a low level of *TLR3* in newborns is a developmentally desirable trend. In a mouse model, Zhang et al. [37] reported higher abortion rates linked with higher *TLR3* levels. It was also mentioned that *TLR3* expression was age-dependent, which can be confirmed by our results. As *TLR3* binds double-stranded RNAs (dsRNAs), its decreased level may increase susceptibility to viral infection in young foals. In our study, the lowest level was recorded before the first suckling, which is in agreement with other reports in premature infants [38] and newborns [39,40]. The data presented above, and the results obtained in our study of the *TLR3* gene, may explain the higher incidence of equine herpesvirus-1 (EHV-1) and equine herpesvirus-4 (EHV-4), responsible for massive respiratory tract infections in foals and young horses. A severe course and high mortality due to contracting equine viral arteritis (EVA) in young horses may also be a result of the decreased expression of *TLR3*. Hussey et al. [41] suggested that *TLR3* plays an essential role in recognizing EHV-1 infections. In our study, a decrease in *TLR3* gene expression was observed after stimulation with Biotropina. We analyzed the level of expression in foals up to 60 days of age, but the literature indicates that the immune system of horses develops most intensively until about 90 days of life [42]. Foals have all of the components of an immune system characteristic of adult horses—but many mechanisms of the immune response have yet to mature. The results indicate that activation of horse monocytes by ligands for the *TLR2* and *TLR4* genes increases their expression, but not that of *TLR3*. Additionally, *TLR3* gene expression decreases with the increase of *TLR4* gene expression after the stimulation of monocytes [43].

### 4.3. Changes in TLR7 Gene Expression

*TLR7* is responsible for recognizing guanidine-rich, single-stranded viral RNA (ssRNA) and is an important mediator of the peripheral immune response. Asquith at al. [41] and Slavica at al. [39] observed no effect of age on*TLR7* gene expression in newborns, similar to Talmadge et al. [22] in horses. Belnoue et al. [44] reported significantly higher levels of *TLR7* gene expression in two-week-old foals compared to adult horses. Harrington et al. [9] found neither an age-dependent pattern in the expression of the *TLR7*/8 genes, nor did they detect the effect of imidazoquinol R848 stimulation on its expression, despite increasing the levels of IL-6 and IL-8. Our results also did not confirm any relationship between a foal’s age and expression of *TLR7*. Until now, little was known about the signaling mechanisms of Toll-like receptors in foals. Identifying the receptors and describing ligands that react with them can provide new insights into immunological responses and can also point to new pathways in the field of therapy and prevention of diseases, particularly infectious ones.

## 5. Conclusions

In summary, on the basis of the results obtained, it was concluded that the expression of the *TLR3* and *TLR4* genes in peripheral blood cells is dependent on age. The expression of the *TLR3* and *TLR4* genes, as well as the levels of immunoglobulins and interleukins, can be modulated by stimulation with the pharmacological agent Biotropina. This experiment demonstrated a strong negative correlation between *TLR3* and *TLR4* gene expression. Detailed knowledge of the molecular mechanisms of immunoglobulin synthesis appears necessary for a better understanding of foal immunity maturity and its influencing factors. At the same time, this experiment encourages studies regarding the influence of the signaling cascade’s proteins on the primary immunological response, providing an opportunity to develop extremely precise methods of regulating acquired immunity. There is still little information about the maturity of a horse’s immune system in the pre- and postnatal period, which negatively affects the planning of health protection strategies for foals.

## Figures and Tables

**Figure 1 animals-10-01966-f001:**
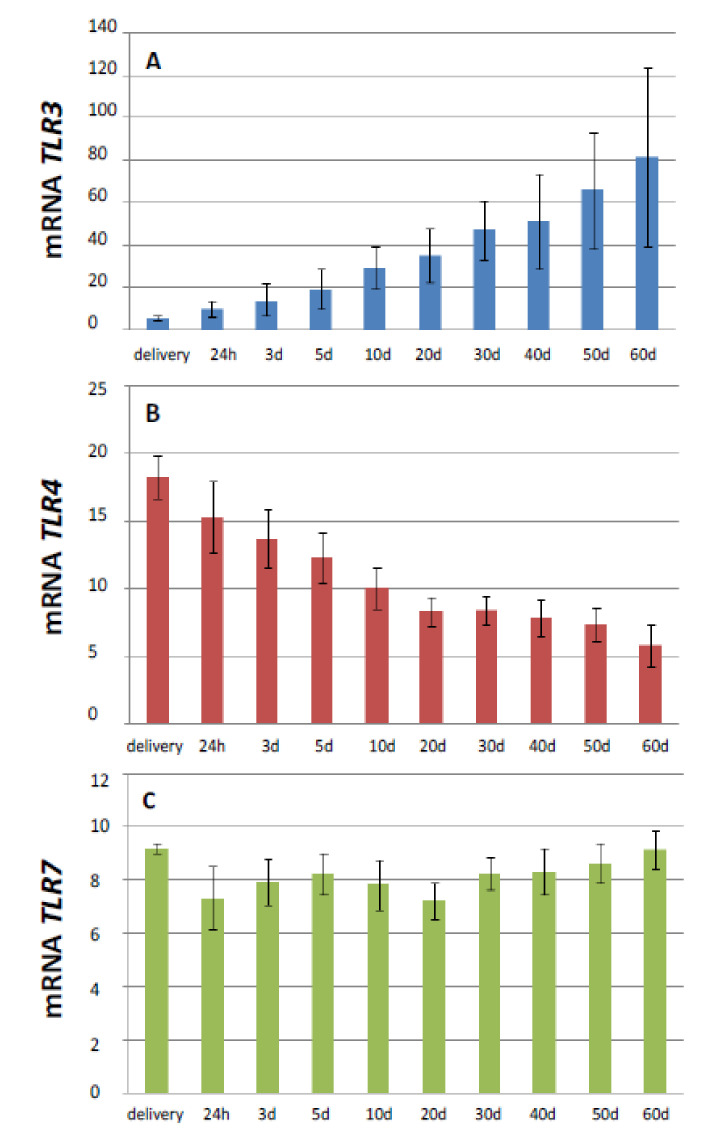
Trends in the change of TLR3 (**A**), TLR4 (**B**), and TLR7 (**C**) expression during foals’ growth. Delivery, sample collected at delivery; 24 h, sample collected 24 h after delivery; 3 days, sample collected every 3rd day after delivery; 5 days, sample collected every 5 days after delivery; 10 days, sample collected every 10 days after delivery; 20 days, sample collected every 20 days after delivery; 30 days, sample collected 30 days after delivery; 40 days, sample collected every 40 days after delivery; 50 days, sample collected every 50 days after delivery; 60 days, sample collected 60 days after delivery. The means are reported with their standard errors.

**Table 1 animals-10-01966-t001:** Probes used for amplification of Toll-like receptor (TLR) genes and housekeeping genes.

Gen.	Full Name of the Gene	Access Number GenBank	TaqMan Gene Expression Assay ID	Dye
*TLR3*	Toll-like receptor 3	NC_009170.2	Ec03467747_m1	FAM
*TLR4*	Toll-like receptor 4	NC_009168.2	Ec03468993_m1	FAM
*TLR7*	Toll-like receptor 7	NC_009175.2	Ec03467530_m1	VIC
*SDHA*	Succinate dehydrogenase complex subunit A	XM_001490889	Ec03470479_m1	VIC
*HPRT*	Hypoxanthinephosphoribosyl transferase	AY372182.1	Ec03470217_m1	VIC

**Table 2 animals-10-01966-t002:** Expression of *TLR3*, *TLR4*, and *TLR7* mRNA over the foals’ subsequent days of age from the control group (mean ± standard error (SE).

Age	*TLR3*	*TLR4*	*TLR7*
Delivery ^1^	6.2 ^A^ ± 0.9 ^2^	18.3 ^A^ ± 2.6	9.2 ± 1.2
24 h	9.8 ^B^ ± 1.7	15.3 ^B^ ± 2.2	7.3 ± 0.9
3 days	14.1 ^A,C^ ± 2.1	13.7 ^C^ ± 1.8	7.9 ± 0.8
5 days	19.1 ^A,B,D^ ± 2.2	12.3 ^A,D^ ± 1.5	8.2 ± 0.9
10 days	29.3 ^A,B,C,E^ ± 2.8	10.1 ^A^ ± 1.1	7.8 ± 0.7
20 days	35.1 ^A,B,C,F^ ± 3.1	8.3 ^A,B^ ± 1.1	7.2 ± 0.6
30 days	48.4 ^A,B,C,D,G^ ± 4.9	8.4 ^A,B^ ± 14	8.3 ± 0.9
40 days	53.6 ^ABCDEF^ ± 5.8	7.8 ^A,B,C^ ± 1.2	8.3 ± 0.7
50 days	80.4 ^ABCDEFG^ ± 8.7	7.3 ^A,B,C^ ± 1.5	8.6 ± 0.7
60 days	87.9 ^A,B,C,D,E,F,G^ ± 8.0	5.8 ^A,B,C,D^ ± 0.9	9.2 ± 07

^1^ Delivery, sample collected at delivery; 24 h, sample collected 24 h after delivery; 3 days, sample collected 3 days after delivery; 5 days, sample collected 5 days after delivery; 10 days, sample collected 10 days after delivery; 20 days, sample collected 20 days after delivery; 30 days, sample collected 30 days after delivery; 40 days, sample collected 40 days after delivery; 50 days, sample collected 50 days after delivery; 60 days, sample collected 60 days after delivery. ^2^ Means are reported with their standard errors. Means with same letter in column show highly statistically significant differences (*p* < 0.01).

**Table 3 animals-10-01966-t003:** Influence of stimulation with Biotropina on the expression of mRNA for selected Toll-like receptors (*TLR3*, *TRL4*, *TLR7*) (mean ± SE).

Age	TLR3		TLR4		TLR7	
	Group C	Group E	Group C	Group E	Group C	Group E
Delivery ^1^	8.3 ** ± 1.8 ^2^	4.1 ** ± 0.55	19.9 * ± 4.0	13.7 * ± 0.5	12.9 * ± 2.9	7.6 * ± 0.9
24 h	14.7 ** ± 3.8	6.5 ** ± 1.24	17.5 * ± 3.3	11.5 * ± 0.6	10.2 ± 2.1	7.0 ± 0.8
3 days	20.0 ** ± 4.9	8.7 ** ± 2.24	16.7 ** ± 2.4	9.9 ** ± 1.1	9.6 ± 1.7	8.3 ± 1.1
5 days	28.0 ± 2.6	15.1 * ± 1.70	14.5 * ± 1.9	9.7 * ± 1.1	11.4 * ± 2.2	7.4 * ± 0.9
10 days	32.7 ± 3.3	30.1 ± 5.43	10.9 ± 0.7	7.1 ± 1.3	9.7 ± 1.4	7.4 ± 0.6
20 days	36.7 ± 4.6	36.4 ± 4.93	8.0 ± 1.2	7.2 ± 1.5	8.7 ± 1.5	7.0 ± 0.7
30 days	62.2 * ± 9.6	39.6 * ± 7.43	8.2 ± 2.3	7.1 ± 1.9	10.0 ± 2.0	7.1 ± 0.9
40 days	78.2 ** ± 6.8	23.1 ** ± 8.82	6.5 * ± 1.3	10.0 * ± 2.4	9.7 ± 1.6	7.9 ± 0.8
50 days	107.0 ** ± 15.3	26.0 ** ± 9.71	5.5 ± 1.9	6.8 ± 1.9	11.2 * ± 1.2	6.9 * ± 0.9
60 days	116.2 ** ± 13.9	44.6 ** ± 10.20	4.2 ± 1.4	6.5 ± 1.7	11.5 ± 1.2	7.6 ± 0.7

^1^ Delivery, sample collected at delivery; 24 h, sample collected 24 h after delivery; 3 days, sample collected 3 days after delivery; 5 days, sample collected 5 days after delivery; 10 days, sample collected 10 days after delivery; 20 days, sample collected 20 days after delivery; 30 days, sample collected 30 days after delivery; 40 days, sample collected 40 days after delivery; 50 days, sample collected 50 days after delivery; 60 days, sample collected 60 days after delivery. ^2^ Means are reported with their standard errors; Group C, control group; Group E, experimental Biotropina-stimulated group (injection at the 35th and 40th days after delivery). * Means in row/line for receptor show significant differences (*p* < 0.05); ** Means in row/line for receptor show highly statistically significant differences (*p* < 0.01).

**Table 4 animals-10-01966-t004:** Correlations (*p*-value) of the expression of *TLR3*, *TLR4,* and *TLR7* mRNA over the subsequent days of age of the foals from the control group.

Gene	Age	*TLR4*	*TLR7*
*TLR3*	<1 h	−0.160 (0.6273)	0.62857 (0.1631)
	24 h	−0.191 (0.4199)	0.462 (0.1400)
	3 days	−0.100 (0.6726)	0.492 (0.1276)
	5 days	−0.095 (0.6912)	0.328 (0.1582)
	10 days	−**0.350 (0.0299)**	0.567 (0.1917)
	20 days	**−0.582 (0.0071)**	0.423 (0.1634)
	30 days	**−0.04361 (0.0085)**	0.368 (0.1098)
	40 days	**−0.56992 (0.0087)**	**0.472 (0.0355)**
	50 days	**−0.46466 (0.0039)**	**0.76541 (0.0251)**
	60 days	**−0.35338 (0.0026)**	**0.82105 (0.0341)**
*TLR4*		<1 h	−0.340 (0.1376)
		24 h	−0.385 (0.0936)
		3 days	−0.472 (0.1355)
		5 days	−0.341 (0.1408)
		10 days	−0.191 (0.4199)
		20 days	−0.319 (0.1707)
		30 days	−0.271 (0.2468)
		40 days	−0.53083 (0.1600)
		50 days	**−0.67519 (0.0111)**
		60 days	**−0.360 (0.0116)**

<1 h, sample collected at delivery; 24 h, sample collected 24 h after delivery; 3 days, sample collected 3 days after delivery; 5 days, sample collected 5 days after delivery; 10 days, sample collected 10 days after delivery; 20 days, sample collected 20 days after delivery; 30 days, sample collected 30 days after delivery; 40 days, sample collected 40 days after delivery; 50 days, sample collected 50 days after delivery; 60 days, sample collected 60 days after delivery. Correlations (*p*-value) bolded show significant differences (*p* < 0.05) while underlined show highly significant differences (*p* < 0.01).

**Table 5 animals-10-01966-t005:** Level of blood morphotic elements in foals (mean ± SE).

	Age	<1 h ^1^	24 h	3 Days	5 Days	10 Days	20 Days	30 Days	40 Days	50 Days	60 Days
Parameters											
Hematocrit (PCV) %	C	50.00 ^2^ ± 1.2	43.33 ± 1.2	39.83 ± 1.5	41.11 ± 1.1	41.56 ± 1.9	34.44 ± 1.6	37.33 ± 1.1	36.33 ± 1.3	37.72 ± 1.2	37.56 ± 1.7
	E	50.50 ± 1.2	44.50 ± 1.8	43.67 ± 1.7	39.58 ± 1.5	41.17 ± 0.9	38.20 ± 1.1	39.75 ± 2.9	36.50 ± 1.7	37.25 ± 1.3	37.75 ± 0.9
Hemoglobin (g/dL)	C	15.46 ± 0.7	14.55 ± 0.7	13.26 ± 0.5	13.75 ± 0.7	13.63 ± 0.7	14.57 * ± 1.2	15.11 * ± 0.9	14.85 * ± 1.3	13.30 ± 0.6	13.98 * ± 1.1
	E	13.94 ± 0.9	13.04 ± 1.7	13.50 ± 0.8	12.71 ± 0.6	12.39 ± 0.2	13.03 ± 0.9	11.80 * ± 0.3	11.70 * ± 0.4	12.39 ± 0.5	10.08 * ± 0.3
RBC count (10^6^/µL)	C	10.93 ± 0.9	10.62 ± 0.7	9.99 ± 0.6	10.55 ± 0.6	9.53 ± 0.4	10.38 ± 1.3	11.24 ± 1.5	9.98 ± 0.8	10.43 ± 0.8	9.86 ± 0.8
	E	11.50 ± 0.5	10.01 ± 0.3	9.17 ± 0.4	10.24 ± 0.9	9.33 ± 0.8	9.04 ± 0.7	9.59 ± 1.0	9.69 ± 0.7	10.29 ± 0.3	10.22 ± 0.3
WBC count (10^3^/µL)	C	7.35 ± 0.8	7.71 ± 0.7	10.35 ± 1.2	9.86 ± 1.0	11.03 ± 0.8	11.30 ± 0.8	12.60 ± 1.0	14.31 ** ± 0.8	14.62 * ± 0.7	13.07 ** ± 0.7
	E	6.25 ± 0.6	8.17 ± 0.5	9.15 ± 0.8	10.64 ± 1.4	12.40 ± 1.1	12.48 ± 0.9	10.91 * ± 1.6	21.75 ** ± 0.5	18.98 * ± 0.4	14.89 ** ± 0.2
Eosinophils (/µL)	C	105 ** ± 2.2	116 ** ± 2.4	72 ** ± 1.5	296 ** ± 6.2	110 ** ± 2.3	226 ** ± 4.7	315 ** ± 6.6	286 ** ± 6.0	292 ** ± 6.1	327 ** ± 6.9
	E	0 ** ± 0.0	16 ** ± 0.3	22 ** ± 0.4	31 ** ± 0.6	37 ** ± 0.7	42 ** ± 0.8	55 ** ± 1.0	206 ** ± 3.7	105 ** ± 1.9	133 ** ± 2.4
Basophils (/µL)	C	44 ± 0.9	39 ± 0.8	31 ± 0.6	99 ± 2.1	55 ± 1.2	113 * ± 2.4	94 * ± 1.9	72 ** ± 1.5	146 ** ± 3.1	196 ** ± 4.1
	E	18 ± 0.3	29 ± 0.5	38 ± 0.7	39 ± 0.7	52 ± 0.9	64 * ± 1.1	75 * ± 1.3	308 ** ± 5.5	205 ** ± 3.7	76 ** ± 1.4
Neutrophils (/µL)	C	4471 * ± 59.5	4488 * ± 59.3	7041 ± 93.4	6903 ± 91.6	5898 * ± 78.3	6104 ± 81.0	7150 * ± 94.9	8583 ± 113.9	8186 ± 108.6	6403 * ± 84.9
	E	5650 ± 50.8	6942 ± 62.5	7374 ± 66.4	7645 ± 68.8	8804 * ± 79.2	7773 ± 69.9	5705 * ± 51.3	9499 * ± 85.5	9109 ± 81.9	7523 ± 67.7
Lymphocytes (/µL)	C	2554 ** ± 35.7	2852 * ± 39.9	3003 * ± 42.0	2465 ± 34.5	4686 ** ± 65.6	4635 * ± 64.9	4945 ** ± 69.2	5150 ** ± 72.1	5701 ** ± 79.8	5880 ** ± 82.3
	E	470 ** ± 4.2	1090 * ± 9.8	1609 ** ± 14.5	2802 ± 25.2	3110 ** ± 28.0	4280 * ± 38.5	4850 ** ± 43.6	10953 ** ± 98.6	9355 ** ± 84.2	7054 ** ± 63.5
Myelocytes (/µL)	C	162 ± 2.3	231 ** ± 3.2	207 * ± 2.9	99 ± 1.4	276 * ± 3.7	226 * ± 3.2	94 ** ± 1.3	215 ** ± 3.0	292 * ± 4.1	261 * ± 3.7
	E	112 ± 1.0	93 **± 0.8	107 *± 0.9	123 ± 1.1	397 * ± 3.6	321 *± 2.9	225 *± 2.0	784 **± 7.1	206 * ± 1.8	104 * ± 0.9

^1^ <1, sample collected at delivery; 24 h, sample collected 24 h after delivery; 3 days, sample collected 3 days after delivery; 5 days, sample collected 5 days after delivery; 10 days, sample collected 10 days after delivery; 20 days, sample collected 20 days after delivery; 30 days, sample collected 30 days after delivery; 40 days, sample collected 40 days after delivery; 50 days, sample collected 50 days after delivery; 60 days, sample collected 60 days after delivery. ^2^ Means are reported with their standard errors. Group C, control group; Group E, experimental Biotropina-stimulated group (injection on the 35th and 40th days after delivery). * Means in row/line for receptor show significant differences (*p* < 0.05); ** means in row/line for receptor show highly statistically significant differences (*p* < 0.01).

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
