# Peer review of "Influence of Age and Immunostimulation on the Level of Toll-Like Receptor Gene (TLR3, 4, and 7) Expression in Foals"

_animals, 2020, doi:10.3390/ani10111966_

Round 1

Reviewer 1 Report

I have no additional concerns about the manuscript.

No other corrections necessary.

Author Response

Dear Reviewer

We would like to thank reviewer for his acceptance of our manuscript and all previous suggestions that helped to improve this manuscript and made it sufficient to be publish in "Animals"

Best regards 

Anna Migdał

Reviewer 2 Report

Understanding age related changes in the neonatal immune response are important for appropriate development of treatment and preventive measures against diseases that are particularly harmful to the very young, be it a horse, a human, or other animal species. While I enjoyed reviewing the initial submission of this paper, I felt that the majority of my previous comments were not appropriately addressed. I commend the authors for addressing the need for further review and addition of other publications regarding similar topics.  Comparison of numerous publications regarding both age related changes in the neonatal immune response and the functionality of TLRs in the neonate strengthened the introduction.

The basic premise as described by the authors was "to provide deeper insight into the molecular mechanisms associated with pathogen recognition by receptors of the innate immune system in foals up to 60 days of age." The experiment was designed to investigate the gene expression of toll-like receptors (TLR), however it overlooked the signaling activity of those receptors. Other investigative measures could have been incorporated to assess the activity of the TLRs in response to various stimulation, thus providing "deeper insight" into the functioning of the neonatal immune response. Understandably, this is difficult to address after an experiment such as this.  It would have been more effective to have included it in the experimental design. 

There are inherent differences between ponies and horses which were not addressed in the discussion, along with several key points regarding the interpretation of the data. I would recommend further evaluation of the results, with an emphasis on strengthening the clarity and depth of the argument.

While this manuscript was fairly well written for submission from a non-native English speaking country, this reviewer recommended review and updating of the manuscript by a native English speaker.  This was not done, as there continue to be numerous grammatical errors throughout the document and supplemental files that will need to be corrected if this manuscript is to be published in English.

Overall, I applaud your efforts to investigate this minimally researched area of equine immunology.  I suggest addressing the issue of the use of a multivalent product as an immunostimulant, along with clarifying the statistics within, and the purpose of, the various tables.  Additional detail should be provided the results and interpretation of those results.

Author Response

Dear Reviewer

below you will find responses to comments.

Understanding age related changes in the neonatal immune response are important for appropriate development of treatment and preventive measures against diseases that are particularly harmful to the very young, be it a horse, a human, or other animal species. While I enjoyed reviewing the initial submission of this paper, I felt that the majority of my previous comments were not appropriately addressed. I commend the authors for addressing the need for further review and addition of other publications regarding similar topics.  Comparison of numerous publications regarding both age related changes in the neonatal immune response and the functionality of TLRs in the neonate strengthened the introduction.

The basic premise as described by the authors was "to provide deeper insight into the molecular mechanisms associated with pathogen recognition by receptors of the innate immune system in foals up to 60 days of age." The experiment was designed to investigate the gene expression of toll-like receptors (TLR), however it overlooked the signaling activity of those receptors. Other investigative measures could have been incorporated to assess the activity of the TLRs in response to various stimulation, thus providing "deeper insight" into the functioning of the neonatal immune response. Understandably, this is difficult to address after an experiment such as this.  It would have been more effective to have included it in the experimental design. 

Aim of this study were rewrite to clearly describe level of performed analysis R.25.

Unfortunately, now it is impossible to add such a analysis into this manuscript but we will try to performed additional study in nearest future

There are inherent differences between ponies and horses which were not addressed in the discussion, along with several key points regarding the interpretation of the data. I would recommend further evaluation of the results, with an emphasis on strengthening the clarity and depth of the argument.

Analysis were performed on horses, therefore in discussion we focused only on this species. Additionally there are lack of available literature  about TLR expression in ponies

While this manuscript was fairly well written for submission from a non-native English speaking country, this reviewer recommended review and updating of the manuscript by a native English speaker.  This was not done, as there continue to be numerous grammatical errors throughout the document and supplemental files that will need to be corrected if this manuscript is to be published in English.

Manuscript will be edited by English editing service provide by MDPI

Overall, I applaud your efforts to investigate this minimally researched area of equine immunology.  I suggest addressing the issue of the use of a multivalent product as an immunostimulant, along with clarifying the statistics within, and the purpose of, the various tables.  Additional detail should be provided the results and interpretation of those results.

In this experiment we wanted to analyse if use of immunostimulant, that is commonly use and have positive effect on animals health (which we know from breeders) like Biotropina influence on expression of TLRs. We did not analyse if this product is working or not - we wanted to know if expression of TLRs is modulated by this immunostimulant and how (increase or decrease level and if any correlation between those TLRs can be found). We are not sure what exactly additional details should be provided because in our opinion our hypothesis that immunostimulant change level of TLRs was proven by all tables we added in this work. It's hard to add additional interpretation based on very little literature data. We can talk only about some trends which may be similar in different animals but in horses there are manuscripts comparing expression from neonatal and adult horses which even are hard to compare with our study. We will be very grateful for information what exactly in statistics and tables should be changed.

Best regards 

Anna Migdał

Reviewer 3 Report

The authors have followed the suggestions. I have some minor formal comments.

  1. 28, write correct abbreviation for microliter.

R.34 …TLR3 and TLR4….

  1. 47 critical
  2. 78 nadve????

Rr. 159, 305, 308, gene abbreviations in italic. R. 159 Changes of.

Discussion section seems to be still too long, but I do not insist on the shortening. Divide it into a few paragraphs.

Author Response

Dear Reviewer

We would like to thank reviewer for his acceptance of our manuscript and all previous suggestions that helped to improve this manuscript and made it sufficient to be publish in "Animals"

  1. 28, write correct abbreviation for microliter.

Abbreviation were changed

  1. 34 …TLR3 and TLR4….

Full names were added

  1. 47 critical

This typing error was corrected

    4. 78 nadve????

We changed this word for commonly use word „naive”

  1. 159, 305, 308, gene abbreviations in italic.

All gene names were italicized

  1. 159 Changes of.

Changed according to suggestion

  1. Discussion section seems to be still too long, but I do not insist on the shortening. Divide it into a few paragraphs.

We divided discssion into three paragraphs R259, 285, 308

  1. Manuscript was edited by Englishediting service provide by MDPI

Best regards

Anna Migdał

Round 2

Reviewer 2 Report

Dear Authors

I appreciate your effort to address my previous comments, especially those that I felt had not been.

The rewording of the aim addressed the concerns regarding deeper insight into the molecular mechanisms, including signaling, that was not initially addressed.  As I stated, I am aware that it is difficult to include that information after the study completion, and I applaud your indication that it will be a topic for future investigation.

Several of my previous comments were still not addressed and can be addressed to improve the manuscript. 

1) Previous comment-There are inherent differences between ponies and horses which were not addressed in the discussion, along with several key points regarding the interpretation of the data. I would recommend further evaluation of the results, with an emphasis on strengthening the clarity and depth of the argument.

Author Response-Analysis were performed on horses, therefore in discussion we focused only on this species. Additionally there are lack of available literature about TLR expression in ponies

Reviewer response- Since information was included in the Materials and Methods regarding the Polish pony breed and similarity to the Tarpan horse (lines 95-97), further discussion of this should be included (line 246-247) where Thoroughbreds were mentioned.  There is enough information for other immunological topics on both ponies and horses, that additional comments can be made.  It will show that the authors did consider those inherent differences.  For example, inclusion of a broad statement, such as “While this study was performed in the Polish pony, the results from this study may differ from potential results if performed in domestic horses” or something similar would improve the discussion and address those concerns.

2) Previous comment-Overall, I applaud your efforts to investigate this minimally researched area of equine immunology. I suggest addressing the issue of the use of a multivalent product as an immunostimulant, along with clarifying the statistics within, and the purpose of, the various tables. Additional detail should be provided the results and interpretation of those results.

Author Response-In this experiment we wanted to analyse if use of immunostimulant, that is commonly use and have positive effect on animals health (which we know from breeders) like Biotropina influence on expression of TLRs. We did not analyse if this product is working or not - we wanted to know if expression of TLRs is modulated by this immunostimulant and how (increase or decrease level and if any correlation between those TLRs can be found).

Reviewer response- It is understood that the authors were not assessing whether or not this product worked.  However, given that this product includes multiple organisms within the product, and each of those classes of organisms are known to affect different classes of toll-like receptors (TLR), the use of such a product needs to have discussion regarding how those may interact and if given separately, cause different responses in the pattern of the TLR response.  This should include the information regarding the specific Gram positive organism in the product and which TLR recognizes it, the specific Gram negative organism included and the TLR expected to recognize it.  A combination of multiple organisms may interact and affect the TLR expression observed in your study.  A broad statement regarding this possibility would strengthen the manuscript.

Author Comment-We are not sure what exactly additional details should be provided because in our opinion our hypothesis that immunostimulant change level of TLRs was proven by all tables we added in this work. It's hard to add additional interpretation based on very little literature data. We will be very grateful for information what exactly in statistics and tables should be changed.

Reviewer response- Additional comments to assist with editing.

Lines 116-119- The differences in group numbers was not addressed.  Please explain why there were 9 animals in the control group and 7 in the treatment group, instead of equal distribution.  Were there more foals expected, but were not included?  What was the reason?

Line 127- Was 5 uL or 5mL used for the injection?  Please verify as ml was used elsewhere in the manuscript to indicate microliters.

Line 142- What was the purpose of using two housekeeping genes?

Lines 156-161- the statistical differences need to be indicated in the figure.

Lines 171-178- Any statistical differences or lack thereof need to be stated.

Line 179- reword “almost on the same level” to “statistically unchanged”

Line 186- update “5until” to “until”

Line 189- Table 3- There are numerous tables included in this manuscript. The information in this table would be better conveyed in a bar graph comparing the control group with the treatment group over time as the x axis, including statistical significance as observed.

Line 245- update “form” to “from”

Supplementary information- also needs English editing.

ThermoScientific information needs to include the state in the USA.

Real-time PCR- As previously mentioned, why were two housekeeping genes used? Please explain.

Figure needs to have statistical significance indicated on the applicable days.  What is the source of this data? Was it only from the control group?  Please indicate all of this information in the figure description, including significance.  These are critical items for the figure.

Author Response

This manuscript is a resubmission of an earlier submission. The following is a list of the peer review reports and author responses from that submission.

Round 1

Reviewer 1 Report

The paper deals with immunostimulation and expression of TLRs and is of interest. Moderate revision is necessary.

You state in Abstract r. 40 and in Results r. 183 “negative correlation” between TLR3 and TLR4 expression, and “no significant correlation” between the age and the expression, rr. 167-8. Similarly Conclusion, r. 280. But where are the correlations? Give them in the table.

MM section, blood sampling and blood analysis:

Have you analysed the blood count and other blood characteristics depending the age and immunostimulation? If yes, give the results in table.

MM section, Animals and feeding:

There is a considerable difference in the age of mares, could it influence the results?

Divide the Results into a few paragraphs.

The Discussion section is too long and a bit tangled. Shorten it to a half of a maximum, and arrange it better.

Formal errors:

140, genes abbreviations in italic.

Reviewer 2 Report

Thank you for the opportunity to review this work.  Understanding age related changes in the neonatal immune response are important for appropriate development of treatment and preventive measures against diseases that are particularly harmful to the very young, be it a horse, a human, or other animal species.  I enjoyed reading this paper, although I felt there were areas needing additional investigation, as well as further review of other publications regarding similar topics. 

The basic premise as described by the authors was  "to provide deeper insight into the molecular mechanisms associated with pathogen recognition by receptors of the innate immune system in foals up to 60 days of age."  The experiment was designed to investigate the gene expression of toll-like receptors (TLR), however it overlooks the signaling activity of those receptors.  Other investigative measures could have been incorporated to assess the activity of the TLRs in response to various stimulation, thus providing "deeper insight" into the functioning of the neonatal immune response. 

There are numerous publications regarding both age related changes in the neonatal immune response and the functionality of TLRs in the neonate, very few of which were cited in this manuscript either in the introduction or in the discussion.  Comparison with these additional publications would strengthen this manuscript, along with performing additional experiments to investigate the functionality of the TLR signaling.  Additionally, there are inherent differences between ponies and horses which need to be addressed in the discussion.

While this manuscript was fairly well written for submission from a non-native English speaking country, this reviewer would recommend review and updating of the manuscript by a native English speaker.  There are numerous grammatical and spelling errors throughout the document that will need to be corrected.  These are numerous and will not be individually identified. 

line 45- “High morbidity and mortality of foals during neonatal period is linked to a physiological lack of immune mechanisms of newborns and the influence of the environment.” Comment-There are publications reporting that immune mechanisms exist in the foal.  The proper stimulus is required for activity. This could be addressed by performing additional experiments for this study.

line 51- “The state of increased susceptibility to infection persists until their passive resistance is achieved.” Comment- Passive immunity is obtained through the transfer of maternal antibodies from the dam to the foal, leading to passive resistance. The use of “passive” in this sentence is misleading and incorrect. The foals’ active immunity must be initiated to achieve resistance.

Line 75- “About 20% of foals die before the end of the second month, which brings significant economic and breeding losses.” Comment- a reference is needed for this statement. Which foals? Where are those foals located? Are there other factors involved in those losses?

Materials and Methods-

Experimental design- Comment- need to address the difference in numbers between the control and treatment groups. There were an even number of foals used in the study, therefore equal numbers in each group would have been ideal.

Line 114-Comment- What were the “controlled conditions” listed in the methods for the experimental group? Were those conditions the same for both groups, except for the administration of the immunostimulant?

Comment- The immunostimulant selected for use contains numerous organisms, both Gram positive and negative with varying PAMPs. With that many PAMPs, the study cannot determine which portion(s) of this treatment is actively stimulating the TLRs. Was the same source used to treat each foal or were there different vials/different lots that could have contained differing concentrations of each organism?

Line 184- “Spearman’s rank correlation test showed strong negative correlation between TLR3 and TLR 4 genes, and lack of correlation between TLR3 and TLR7, TLR4 and TLR7.” Suggest addition of “as well as” into-‘correlation between TLR3 and TLR7, as well as, TLR4 and TLR7’ to improve clarity.

Line 187- “Available literature data indicate that the influence of age on TLR4 expression in horses is doubtful.” Comment- Additional references should be cited, as there is more available information. This section should have further clarification of the methods used and findings by other studies cited, as there seems to be some discrepancy as to the claims.

“Vendrig et al. found no differences between the expression of TLR4 in blood mononuclear cells of foals at 12h of life or in adult horses [8]. Stimulation with lipopolisacharide (LPS) resulted in higher TLR4 mRNA expression in adult horses, while no response to LPS stimulation was found in foals.” Comment- this needs a reference or if it is from the previously cited article, the reference needs to be moved to the end to clarify. Additionally, further clarification is needed, as it infers that the LPS stimulation occurred directly in the horse, when it is more likely to have occurred in mononuclear cell culture.

Line 196- “Moreover, LPS stimulation resulted in higher level of TLR4 mRNA expression, while no response was found for PGN stimulation (PGN – peptidoglycan from Staphylococcus aureus).” Comment- The discussion of TLR function is not appropriate given the study described in this manuscript does not report findings of functionality. If additional experiments are performed, this discussion would be more appropriate.

Line 199 and on- Comment- This and the following paragraphs discuss numerous studies in various species, however, the types of studies performed have significantly different purposes and methods. To more accurately compare findings, more similar studies should be used for those comparisons. As above, if additional experiments are performed to expand on the functionality of the TLRs, this discussion would be more appropriate.

Figures and Tables-

Data in tables and graph could be better demonstrated in 3 separate bar graphs, one for each TLR, comparing the results for the control group to the treatment group.

In summary, significant work needs to be done to improve this manuscript, so that it will provide the "deeper understanding" that was the goal of the study.  With incorporation of the recommended changes, this manuscript could be a useful addition to the body of information already in publication.

Reviewer 3 Report

While this paper provides some information regarding TLR expression in foals and the effect of treatment with an immune modulator, the authors fail to provide a convincing argument as to its importance.  The introduction ignores most of the pertinent equine literature, relying primarily upon a single reference.  Had they reviewed this literature they would have noted that others have addressed the issue of TLR responsiveness in foals noting that foals are hyporesponsive to this stimulation up to 30 days of age and then begin to show adult-type responsiveness thereafter.  As such, it would have been more interesting, and pertinent, had they treated the foals in this study with the immune modulator earlier.  The introduction also contains uncited statements of questionable accuracy (e.g. "20% of foals die before the end of the second month").  I do not know where they derived this information as it certainly does not reflect what is already known and widely published.